# Human populations in the world's mountains: Spatio-temporal patterns and potential controls

James M. Thornton[1]*, Mark A. Snethlage[2], Roger Sayre[3], Davnah R. Urbach[2], Daniel Viviroli[4], Daniele Ehrlich[5], Veruska Muccione[4], Philippus Wester[6], Gregory Insarov[7], Carolina Adler[1]

**1** Mountain Research Initiative, University of Bern, Bern, Switzerland, **2** Global Mountain Biodiversity Assessment, University of Bern, Bern, Switzerland, **3** U.S. Geological Survey, Reston, VA, United States of America, **4** Department of Geography, University of Zurich, Zurich, Switzerland, **5** Joint Research Center, European Commission, Ispra, Italy, **6** International Centre for Integrated Mountain Development (ICIMOD), Kathmandu, Nepal, **7** Institute of Geography, Russian Academy of Sciences, Moscow, Russia

* james.thornton@unibe.ch

**Data Availability Statement:** All files are available from https://zenodo.org/record/6673651 or, in the cases of certain large datasets, via links contained in the manuscript / SI.

## Abstract

Changing climate and human demographics in the world's mountains will have increasingly profound environmental and societal consequences across all elevations. Quantifying current human populations in and near mountains is crucial to ensure that any interventions in these complex social-ecological systems are appropriately resourced, and that valuable ecosystems are effectively protected. However, comprehensive and reproducible analyses on this subject are lacking. Here, we develop and implement an open workflow to quantify the sensitivity of mountain population estimates over recent decades, both globally and for several sets of relevant reporting regions, to alternative input dataset combinations. Relationships between mean population density and several potential environmental covariates are also explored across elevational bands within individual mountain regions (i.e. "sub-mountain range scale"). Globally, mountain population estimates vary greatly—from 0.344 billion (<5% of the corresponding global total) to 2.289 billion (>31%) in 2015. A more detailed analysis using one of the population datasets (GHS-POP) revealed that in ~35% of mountain sub-regions, population increased at least twofold over the 40-year period 1975–2015. The urban proportion of the total mountain population in 2015 ranged from 6% to 39%, depending on the combination of population and urban extent datasets used. At sub-mountain range scale, population density was found to be more strongly associated with climatic than with topographic and protected-area variables, and these relationships appear to have strengthened slightly over time. Such insights may contribute to improved predictions of future mountain population distributions under scenarios of future climatic and demographic change. Overall, our work emphasizes that irrespective of data choices, substantial human populations are likely to be directly affected by—and themselves affect—mountainous environmental and ecological change. It thereby further underlines the urgency with which the multitudinous challenges concerning the interactions between mountain climate and human societies under change must be tackled.

**Funding:** For the period 2020-2021, in which the analyses were conducted and the manuscript prepared and submitted, J.M.T, C.A., and P.W. acknowledge funding from the Swiss Agency for Development and Cooperation (SDC; https://www.eda.admin.ch/sdc) (Project number: 7F-10208.01.02). C.A. also acknowledges additional support from the SDC (Project number: 7F-10240.01.01) and the Swiss Academy of Sciences (SCNAT; https://scnat.ch/en) via its support for the Mountain Research Initiative (Project number: FNW0004_004-2019-00). G.I. acknowledges funding from the Institute of Geography, Russian Academy of Sciences (http://www.igras.ru/en/node/1) (Project number: AAAA-A19-119021990093-8 (FMGE-2019-0007)). The sponsors played no direct role in the study design, data collection and analysis, decision to publish, or preparation of the manuscript.

**Competing interests:** The authors have declared that no competing interests exist.

# Introduction

Humanity is highly dependent on mountain ecosystem goods and services. However, climate change has already had major impacts upon the environmental and ecological systems embedded in the world's mountains [1]. As such, geographically widespread ongoing (and in many cases accelerating) responses of these natural systems will almost certainly bring about numerous adverse societal consequences [2], although there may also be some opportunities. For instance, the seasonal snowpack has traditionally represented a vital store and delayed source of freshwater in many regions [3, 4], yet snow parameters are now changing—with declines evident in many regions [5, 6]. Similarly, the continued retreat and diminution of mountain glaciers will substantially alter the flow dynamics of many large ice melt-dominated rivers, jeopardizing the water resources that sustain large populations –- notably in and downstream of the Tropical Andes, the Hindu-Kush Himalayas, and the mountains of Central Asia [7]. Some cryosphere-driven changes in streamflow dynamics are already detectable [8–10].

Other aspects of mountain environments and ecosystems are likewise responding to changing conditions. For example, species and habitats on mountain summits and across elevational gradients are redistributing [11–13] and mountain permafrost is degrading [14–16]. Given the highly interconnected nature of mountain systems, changes in individual components can propagate widely, sometimes initiating complex feedback mechanisms [17]. Amongst other consequences, the frequencies and magnitudes of floods, landslides, rockfalls, glacial lake outburst floods, forest fires, and snow avalanches are thought to be increasing in many regions [18–21], although some evidence is contrasting or remains lacking.

Simultaneously, human populations in and near many mountainous regions are changing and redistributing in response to various context-specific drivers—social, economic, political, and environmental insecurity amongst them [22]. In areas experiencing strong population growth and urbanization, pressures on mountain ecosystem services (e.g. demand for water, land, and recreation opportunities) are naturally exacerbated. For example, the dependence of lowland populations on mountain water resources continues to increase [23]. In addition, although humans have actually modified much of the natural environment to some degree for thousands of years [24], more recent intensified unsustainable land-use practices and over-exploitation of natural resources associated with population and economic growth have caused the disappearance, fragmentation, and degradation of many mountain habitats [25–29]. Conversely, regional declines in mountain populations, often due to the abandonment of pastoralism and other traditional practices, can negatively impact mountain cultural landscapes, ecosystems, and economies—and hence remaining inhabitants. Any such trends would be concerning because even at present, many mountain people struggle to meet their basic water, energy, and food needs [30].

Mountains and their adjacent and/or dependent regions therefore clearly contain complex socio-environmental and ecological systems. As such, research, policy, and practical efforts must span the interface between the changing provision of mountain ecosystem services, natural hazards propensities, and other bio-physical phenomena on the one hand, and the changing human populations and societies who are variously exposed to, rely upon, and (over) exploit them on the other. Unfortunately, a chasm often exists between the quantity, quality, and availability of data and information across this thematic divide, with socio-economic data —of which estimates of the number of people living in and near mountains (henceforth "mountain population") are just one (important) element—often being severely limited. For example, with respect to global flood risk, Ward et al. [31] stated that "obtaining detailed information on exposure and vulnerability at the global scale remains an open challenge", and "without profoundly improved representations of exposure and vulnerability, gains from

improved hazard modelling will not filter through to improved risk estimates". Consequently, predictions of precisely how localised physical changes and events in mountains, which are themselves frequently difficult to foresee, will propagate through their broader socio-environmental/ecological systems often remain highly uncertain.

Estimating mountain population requires both spatially explicit (i.e. gridded) population count data and means for delineating mountain regions. Several such population data products, developed using various approaches, provide estimates over recent decades up to the (near) present and are freely available [32]. Similarly, mountainous areas have been delineated from several perspectives, including via topographic characterization and according to climatic characteristics, socio-political and/or cultural constructs, and indigenous knowledge. However, even using perhaps the most conceptually simple, topography-based approach, differences in both input datasets and the essentially arbitrary and empirical algorithmic choices that are required to determine which precise terrain conditions should be considered "mountainous" lead to several contrasting outcomes. Indeed, the three global mountain delineation approaches—"K1" [33], "K2" [34], and "K3" [35]—that are most prominent and widely applied differ greatly in the proportion of total land surface area that they consider to be mountainous [36, 37]. More regional mountain delineation efforts typically involve similar approaches [38, 39]. To generate urban population estimates, urban extent boundaries are naturally also required, and again alternative delineation approaches exist in this regard.

Several previous studies [37, 40–46] have combined such datasets to seek to tackle the rather fundamental questions of *how many people inhabit the world's mountainous or "high altitude" (strictly, "elevation" [47]) regions?* and, more generally, *how is the global population distributed hypsometrically?* However, none has yet done so in a truly comparative fashion. In other words, the impacts of employing the various alternative, often "equally valid" input datasets on the resultant estimated remains to be comprehensively and rigorously assessed [45]. Equally importantly, previous studies have not consistently achieved methodological transparency and reproducibility, and regional and urban mountain population dynamics have received much less attention.

Consequently, many users of mountain population estimates—who cannot necessarily be expected to possess the technical background required to fully appreciate the subtleties and complexities involved—may have struggled to decipher precisely how the results they relied upon were computed, and the extent to which employing alternative input datasets might have produced different outcomes across a range of scales from global to regional. Such a situation is unconducive to equitable and defensible policy-making and subsequent actions, especially if mountain population estimates do indeed demonstrate high sensitivity to input dataset choices—as one may expect given the contrasts in global mountain delineation extents.

A need also exists to progress beyond purely descriptive statistics and explore any relationships that may exist between mountain population and potential environmental and other covariates. Several global-scale studies and reviews along these lines have been conducted with respect to both pre-agricultural/industrial and industrialised populations [48–51], although the effects of potential environmental drivers on human abundances remain incompletely known [50]. None of these studies were mountain specific, however (but see [52] with respect to hunter-gatherers in the Rocky Mountains).

To our best knowledge, there have been no quantitative attempts to explore the possible environmental influences on (contemporary) mountain populations explicitly. Elucidating the nature of any such relationships in mountains could lead to improved understanding of some of the underlying drivers of population dynamics, including the identification of topographic, climatic, and other conditions under which high population densities can develop and be sustained in these regions. In turn, such knowledge could help evaluate current and improve

future spatially explicit mountain population projections, which could again ultimately contribute to more robust forward-looking decision making.

In this context, a script-based workflow that relies exclusively on open data and software is first developed (see the online Supplementary Information; S1 File). We then apply the workflow to demonstrate that it is capable of efficiently addressing a wide range of outstanding questions, such as:

1. To what extent do estimates of the global human population living in and around mountains (i.e. the "mountain population") depend on input data choices?

2. How have mountain population counts and densities varied spatially and temporally over recent decades?

3. How do population density estimates in mountains compare with those of their wider regions?

4. Which mountainous regions are undergoing the most profound population changes?

5. What proportion of the mountain population can be considered "urban", and to what extent are recent population change and urban extent change in mountains spatially related?

6. To what extent are mountain population densities within individual mountain regions related to topographic, climatic, and protected-area variables, and how have these dependencies changed in time and space over recent decades?

## Materials and methods

### Data: Selection and preparation

The method employed entirely open-source datasets and software tools. The datasets used (see also S1–S5 Tables in S1 File) can be grouped into the following categories: i) mountain delineations, ii) population grids, iii) aggregation/reporting polygons, iv) urban area extents, and v) other potential population covariate datasets (e.g. topographic, climatic, and protected-area extent layers). These datasets are briefly described below.

Three alternative binary raster delineations of the world's mountainous regions, each of which rely on different terrain datasets and methodological choices, and were originally developed with somewhat different applications in mind, were considered; here, these layers are referred to as "K1" [33], "K2" [34], and "K3" [35] layers (S1 Table in S1 File; see S1–S3 Figs in S1 File). In K1, which was developed in order to map global mountain forests, pixels are classified as mountainous or not according to whether the combination of their elevation, slope, and ruggedness (or relative relief) values exceed certain predefined thresholds. In K2, which was developed to facilitate mountain biodiversity comparisons, a similar approach was taken, except ruggedness was the sole criterion. Meanwhile, in K3, which was developed in the course of a global ecosystem mapping exercise, mountains were delineated by extracting this specific class from several so-called "ecological land units". Sayre et al. [36] provide further details on these three delineations.

Particular care was taken here to ensure that only the authoritative, author- and institution-sanctioned versions of these layers were employed; as a secondary outcome of this project, these layers have been released on the USGS's Global Mountain Explorer (GME) v2. In particular, the K2 layer used represents a substantial update on that described by Körner et al. [34] and applied by Körner et al. [43] in the sense that to generate the binary map, no resampling of the underlying 30 arc-second (0.00833˚) ruggedness grid was undertaken. As such, much

more information content is retained in this version. In K1, mountainous parts of Antarctica and surrounding islands, as well as "Class 7" ("isolated inner areas") everywhere, were removed prior to analysis. The K1 layer used does however retain a large amount of mountainous terrain at high elevations of the Greenland Ice Sheet, which will affect some population density results presented.

Several gridded population datasets, which in addition to having been developed using different methods are of various spatial resolution, were considered (S2 Table in S1 File). All sources provide multi-year estimates over recent decades. However, only some support temporal analyses (i.e. can be applied to investigate changes through time on a comparable basis); in other cases (e.g. LandScan), the metadata provided indicates that they should not be used in this way. In these cases, only the most recent year was used. 2015 was identified as a year common to all datasets that could be used as a basis for their comparison. HYDE3.2 [53] represented another potential population data source, but given its intended purpose of quantifying human population dynamics over the entire Holocene, and its rather coarse spatial resolution (0.0833˚), we elected to omit it from our analysis. A thorough review of global-scale gridded population datasets is provided by Leyk et al. [32]. It must be noted that these datasets may partially overlook individuals in conflict zones and nomadic and/or pastoralist populations.

Two alternative data sources for characterising global urban extents over recent decades were employed (S3 Table in S1 File). The GHS Settlement Model grid (GHS-SMOD [54]) identifies urban areas on the basis of GHS-POP (one of the population datasets used). Only pixels corresponding to "urban centres" (code = 30) were treated as urban in our analysis. These areas are comprised of contiguous 1 $km^2$ cells with densities of $\geq$1,500 inhabitants/$km^2$ and a total population $\geq$50,000. Gaps in these urban centres had already been filled and their edges smoothed in the incoming data [55]. The second data source takes a contrasting approach; Global Urban Boundaries (GUB) are automatically delineated from artificial impervious area data [56].

The aggregation polygon datasets (S4 Table in S1 File; S4–S8 Figs in S1 File) were selected such that population estimates attributed to them are likely to be relevant for various common reporting needs and other applications. For instance, regional boundary sets used in the Intergovernmental Panel on Climate Change (IPCC) and the Intergovernmental Science-Policy Platform on Biodiversity and Ecosystem Services (IPBES) assessments were incorporated. Here, we used the IPCC AR6 Working Group II regions, which are very similar to those defined by Hewitson et al. [57]. The latest version of the Global Mountain Biodiversity Assessment's (GMBA's) Mountain Inventory (v2) [58, 59], which provides named mountain range extent polygons within a hierarchical system, was also used for aggregation purposes. In the specific incoming GMBA layer used, external boundaries were buffered beyond the maximum combined extent of K1, K2, and K3 by approximately 5 km. As such, population estimates for the GMBA entire regions include adjacent populations falling within this buffer, whilst the corresponding "GMBA mountain-only" estimates pertain exclusively to populations within both the polygons and the respective K1, K2, or K3 extents. The World Climate Regions (WCRs) of Sayre et al. [60] were incorporated to facilitate a preliminary assessment of the large-scale association between human population density and climate to be undertaken, both globally and specifically within mountains. Finally, BasinALTLAS [61] is a hierarchical dataset of hydrological (surficial) drainage basin outlines. The Level 6 polygons were selected for use here so that some downstream non-mountainous areas were encompassed. Therefore, the BasinALTLAS "entire region" population estimates also provide some insight into the number of inhabitants living in the hydrologically connected vicinity of mountains. Through hydrological flows in particular, mountains of course also sustain far more distant populations in many parts of the world [7, 23], but assessing this dependence is not the primary objective of this

study. The input polygon datasets used are provided in the online S1 File. For some layers, pre-processing was necessary to ensure full geometrical validity.

The additional datasets that were introduced for the exploratory part of the analysis are summarized in S5 Table in S1 File. Pre-processing of the World Database on Protected Areas (WDPA [62]) was necessary. Specifically, all protected-area polygons either designated as UNESCO-MAB Biosphere Reserves, with statuses "Proposed" or "Not reported", or with the attribute "Marine" = 2 (i.e. predominantly or entirely marine) were removed. The remaining features were then dissolved to remove any overlapping geometries (see [62] for further details). Protected areas that were only represented as points were also omitted on account of the ambiguity associated with their actual protected extents.

### Processing workflow overview

Although the workflow is conceptually rather simple, its implementation for such a large combination of (large) databases was not trivial. Given the size of certain input datasets and the extensive and efficient geospatial functionality required, PostGIS—an extension that adds spatial functionality to the widely used, open source PostgreSQL object-relational database system—was identified as the most appropriate main analysis software. Here, PostgreSQL 13.1, PostGIS 3.1, and PgAdmin 4 v5.0 were used alongside GDAL 3.0.4, QGIS 3.14.1, and R 4.0.4. To avoid the compromises inherent to global-scale projections, all layers retained their original geographic coordinates (EPSG:4326) for all spatial processing and analysis steps. All areas were calculated using PostGIS's "Geography" data type, i.e. on the spheroid.

The workflow can be separated into three somewhat distinct components (S16–S18 Figs in S1 File). Part A consists of a comprehensive analysis to derive various area, population, and population density estimates for all considered combinations of mountain delineations, population grids, and years. Results are reported both at the global level and against the five different regional aggregation/reporting polygon datasets. For the regional component, population sums and densities were computed within both the entirety of each zone (or region) and within only the corresponding mountainous portions for all possible combinations of mountain delineations, population grids, and reporting boundary datasets. Mean mountain population densities within each zone of the various reporting datasets could therefore be compared with those across the entire corresponding zones. Part B is concerned with the quantification of urban mountain populations, and involved only a subset of the population grids used in Part A. For internal consistency, results were not generated for combinations of urban extents and population grids that did not correspond to exactly the same year.

Part C sought to explore the patterns and dependencies between mountain population density / urbanization estimates and several potential environmental covariates summarised across three elevation bands within each mountain region, which we refer to as sub-mountain range scale. To keep the workflow manageable, the only population and urban extent datasets involved in this phase were GHS-POP and GHS-SMOD. We are confident that these datasets provide representative results, however. The mountainous parts of the GMBA polygons (defined according to their intersection with the single geometrical union of K1, K2, and K3 extents) were first divided into one of three zones—"Low", "Middle", and "High"—according to the 0.33 and 0.67 quantiles of the elevation distribution within each. S10 Fig in S1 File illustrates the outcome of spatial sub-division across the Himalayan and Central Asian region. Then, for each resultant zone, population density and urbanization metrics were then derived from GHS-POP and GHS-SMOD, respectively, before potential covariate data such as the protected proportion and zonal means of the various topographical and climatic layers were attributed. To investigate the extent to which projected future mountain population metrics

might be related to topographic and (present day) climatic and protected-area variables, additional correlation matrices (both the simplified and more detailed version) were generated by substituting population metrics derived from the five (global) Shared Socioeconomic Pathways (SSPs) for the years 2050 and 2080.

Variation in cell size with latitude was not accounted for when calculating the covariate means across the respective GMBA sub-zones (i.e. all cells within a given zone were equally weighted). This may induce a slight bias in estimates for zones that span large latitudinal ranges, but even in such cases impacts are expected to be minor and unlikely to affect the conclusions drawn. The resultant tables are provided in spatial format in the online S1 File (*gmba_v2_0_k_union_diss_contours_covariates_joined.sqlite* and *gmba_v2_0_k_union_diss_contours_covar_joined_for_ matrix_plt.sqlite*).

Scatter plots of population density against all potential covariates were then produced (see the online S1 File), and the correlation between all variable pairs quantified using Spearman's $\rho$. This non-parametric coefficient was an appropriate choice because the univariate distributions failed Shapiro-Wilk tests for normality.

For full algorithmic details, see the scripts provided. The Greenwich Equal Earth projection (EPSG: 8857) was applied to generate the output maps. To replicate the workflow, a powerful machine with high RAM and a large, fast disk (ideally SSD) would be beneficial. PostgreSQL configuration settings should be set according to the hardware available. We used a Windows workstation (Intel® Xeon® W-2255 CPU @ 3.70GHz, 10 cores/20 logical processors, 128GB RAM).

## Results and discussion

### Global estimates

We calculated that K1, K2, and K3 cover 24, 13, and 30% of the global land surface area (excluding Antarctica), respectively (see S1 and S6 Tables in S1 File). Fig 1 shows the evolution of global total (i.e. mountain and non-mountain) and (global) mountain-only population over recent decades according to the various mountain delineations and gridded population datasets employed. Both the global and mountain-only populations are seen to have increased linearly with time. The agreement between the respective global estimates for 2015—the only year for which all population sources provide data—is very high, although this is likely the case for other years too. Beneath the global totals, the mountain-only estimates form three clear groups. These groups correspond to the choice of mountain delineation. There is also variability within each group, which reflects the choice of population dataset.

Global estimates of the mountain population in 2015 thus vary considerably from 0.344 billion to 2.289 billion; a range equating to <5% to >31% of the corresponding global totals. The choice of mountain delineation is clearly the principal factor driving this variability, with the contrasting extent to which each delineation includes or excludes very populous cities located in or near mountainous terrain likely playing a key role. The effect of the population data source is evidently discernible but secondary. Variability in the population estimates for a given year and mountain delineation are likely driven by a combination of the different population grid input datasets and generation methods, as well as their contrasting spatial resolutions. All else being equal, and assuming macro-scale spatial population distribution to be somewhat non-uniform, the contrasting spatial resolutions of these datasets alone can affect the mountain population statistics by altering the number of pixels that fall within the mountainous extents.

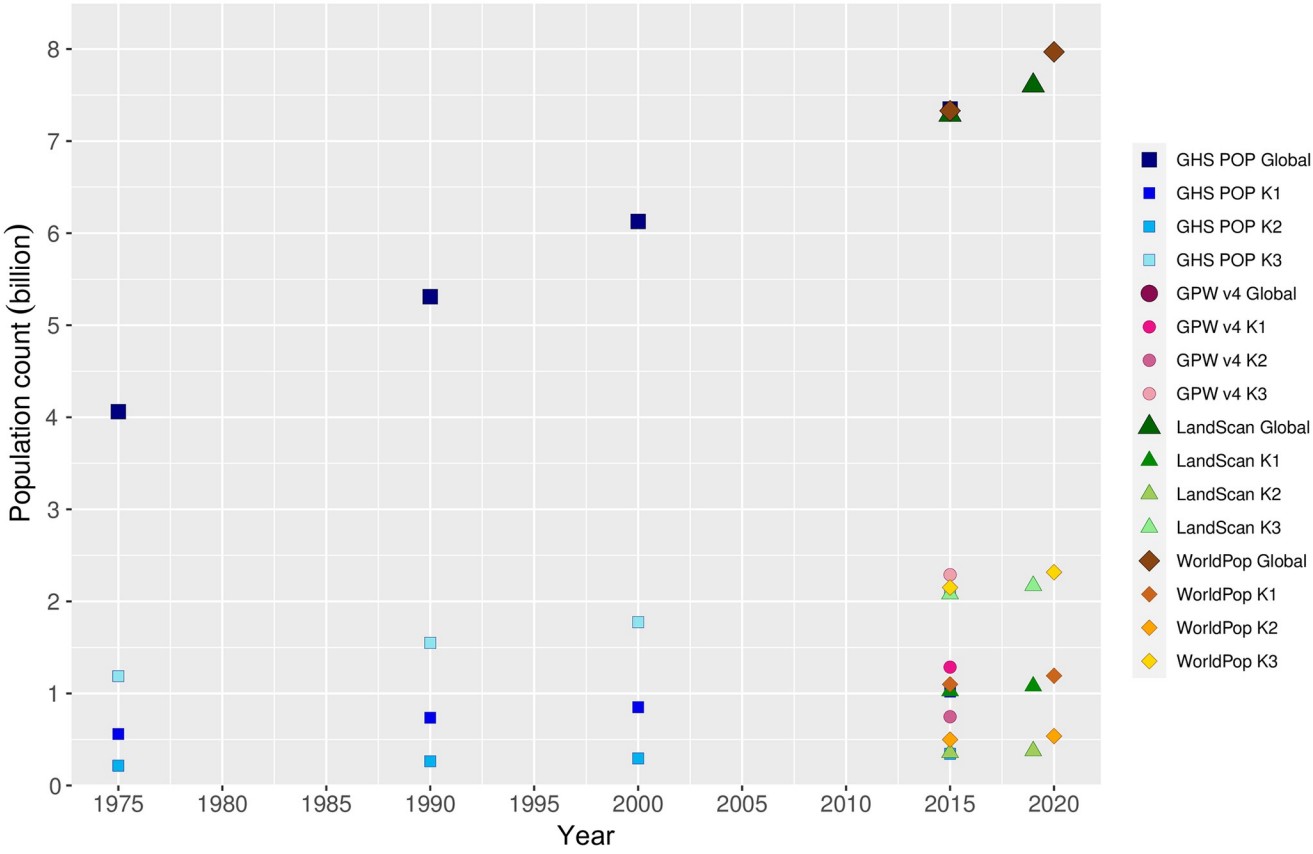

**Fig 1. Global total (circles) and mountain-only (triangles) population over recent decades according to all input population and mountain delineation datasets combinations.** Note that some of the data points for the 2015 estimates are partially obscured by the others. The data plotted are presented in S7 Table in S1 File.

## Urbanization and population change

Table 1 presents our global total and mountain-only urban population estimates according to several combinations of urban and mountain extent delineations. As mentioned above, the two urban extent sources considered, GHS-SOMD and GUB, were derived using different methods (namely a population density-based approach and a land surface type-based approach, respectively). For this analysis, a reduced set of population grids was considered; GHS-POP was used primarily, although WorldPop—which has higher spatial resolution—was also introduced to provide comparisons for 2015.

Whilst urban areas according to both GHS-SMOD and GUB cover only a very small proportion of the total global land surface, they account for over 30% of the current (2015) global population. The proportion of the total global urban population that can be considered mountainous spans a broad range; for 2015, from as little as ∼1.4% to as much as ∼34% depending on the input dataset combination used (of those considered in this phase). Combining Table 1 and S7 Table in S1 File, meanwhile, reveals that the proportion of the total mountain population living in urban areas in 2015 ranged from 6% to 39% (again according to the dataset combinations considered here). The broad range in these estimates makes conceptual sense, since many large and populous cities are situated close to—or in the foothills of—mountains. On Taiwan and the east coast of China, for instance (S9 Fig in S1 File), little (GUB) urban extent falls within K2, but substantially more falls within K3. Table 1 also reveals that employing the

**Table 1. Summary of global total and mountain-only urban population estimates according to GHS-POP and (for 2015) WorldPop population datasets for alternative combinations of urban and mountain extent delineations.**

| Year | Urban extent delineation | Population data source | Global urban area | | Global urban population | | Mountain urban population | | | | | |
|---|---|---|---|---|---|---|---|---|---|---|---|---|
| | | | | | | | K1 | | K2 | | K3 | |
| | | | km$^2$ | % of total land surface area (exc. Antarctica) | Sum | % of global population | Sum | % of global urban sum | Sum | % of global urban sum | Sum | % of global urban sum |
| 1975 | GHS-SMOD | GHS-POP | 305,391 | 0.23 | 1,504,875,604 | 37.05 | 136,293,337 | 9.06 | 43,603,453 | 2.90 | 366,753,770 | 24.37 |
| 1990 | GHS-SMOD | GHS-POP | 428,016 | 0.32 | 2,196,689,230 | 41.37 | 204,826,903 | 9.32 | 49,879,819 | 2.27 | 524,838,068 | 23.89 |
| 2000 | GHS-SMOD | GHS-POP | 531,457 | 0.39 | 2,704,125,652 | 44.14 | 258,562,671 | 9.56 | 56,010,690 | 2.07 | 638,930,475 | 23.63 |
| 2015 | GHS-SMOD | GHS-POP | 663,545 | 0.49 | 3,522,599,171 | 47.93 | 350,774,246 | **9.96** | 75,052,035 | **2.13** | 825,284,963 | **23.43** |
| 2015 | GHS-SMOD | WorldPop | 663,545 | 0.49 | 2,545,311,972 | 34.63 | 210,918,122 | **8.29** | 47,621,189 | **1.87** | 584,086,040 | **22.95** |
| 1990 | GUB | GHS-POP | 300,345 | 0.22 | 1,255,699,640 | 30.92 | 98,498,672 | 7.84 | 17,741,278 | 1.41 | 276,327,171 | 22.01 |
| 2000 | GUB | GHS-POP | 447,880 | 0.33 | 1,905,271,214 | 35.88 | 159,727,391 | 8.38 | 27,357,503 | 1.44 | 434,054,652 | 22.78 |
| 2010 | GUB | N/A | 590,132 | 0.44 | N/A | N/A | N/A | N/A | N/A | N/A | N/A | N/A |
| 2015 | GUB | GHS-POP | 636,568 | 0.47 | 2,693,086,812 | 36.64 | 227,651,552 | 8.45 | 38,315,433 | 1.42 | 611,906,762 | 22.72 |
| 2015 | GUB | WorldPop | 636,568 | 0.47 | 2,204,922,828 | 30.00 | 169,465,867 | **7.69** | 30,359,802 | **1.38** | 496,962,506 | **22.54** |
| 2018 | GUB | N/A | 809,366 | 0.60 | N/A | N/A | N/A | N/A | N/A | N/A | N/A | N/A |

Urban population estimates for 2010 and 2018 are omitted because GHS-POP does not provide data for these years.

higher-resolution WorldPop dataset leads to systematically lower urban population estimates, all else remaining constant. This could indicate that use of coarse population grids in conjunction with smaller reporting areas (such as urban areas) may introduce systematic bias, although methodological differences in the generation of the dataset may also partly explain this observation.

Fig 2 illustrates the increase in extent and population of a well-known "mountain city" (i.e. a city that is considered at least partially mountainous by all three delineations)—Santiago, Chile—according to the GHS-POP population grids and the Global Urban Area (GUB) extents. The global-scale spatial outputs from which this plot is generated, i.e. population metrics attributed to all urban extent polygons globally for the years considered, are provided in the online S1 File.

Fig 3 meanwhile shows the bivariate relationship between population count (i.e. sum) change and urban extent change by GMBA mountain range sub-zone over the period 1975 to 2015 according to the GHS-POP and GHS-SMOD datasets.

Fig 3 enables one (for instance) to differentiate between those mountain regions where population growth is accompanied by strong urbanization and those where it remains mostly rural. Such distinctions are crucial for applications such as environmental management and sustainable development planning. Whilst the spatial patterns in relationship between mountain population change and urban extent change are somewhat complex, in the mountains of Central America, East Africa, the Middle East, and parts of South-East Asia in particular, population growth over the period in question has clearly been accompanied by strong urbanization. Conversely, population growth regions such as the northern Rockies and Alaska, Scandinavia, the Tibetan Plateau, and eastern Eurasia has been largely non-urban (i.e. rural or suburban). Regions where urban extents have increased but population densities have remained fairly constant or decreased are noticeably less widespread. Mountain population declines are evident in some regions (e.g. much of Italy). The similar bivariate choropleth map showing population change against urban population change (rather than urban extent

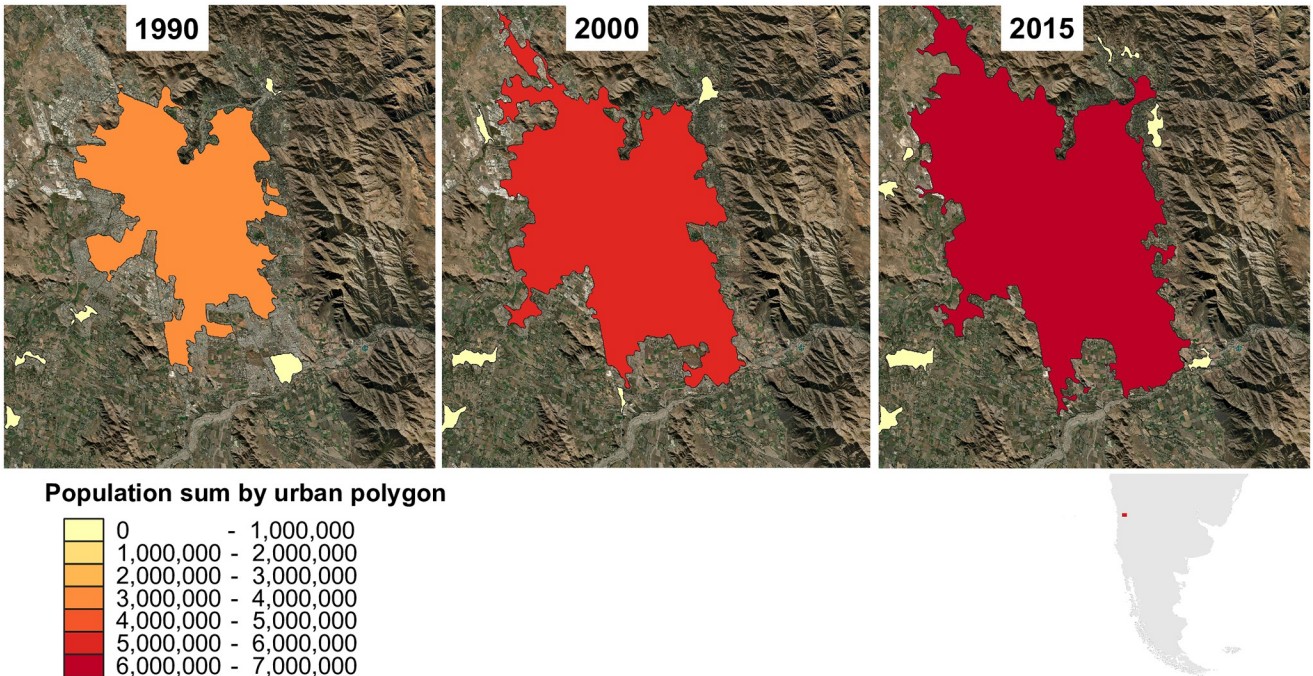

**Fig 2. Growth in extent and population of Santiago, Chile, over recent decades according to GHS-POP population data and GUB urban extents.**

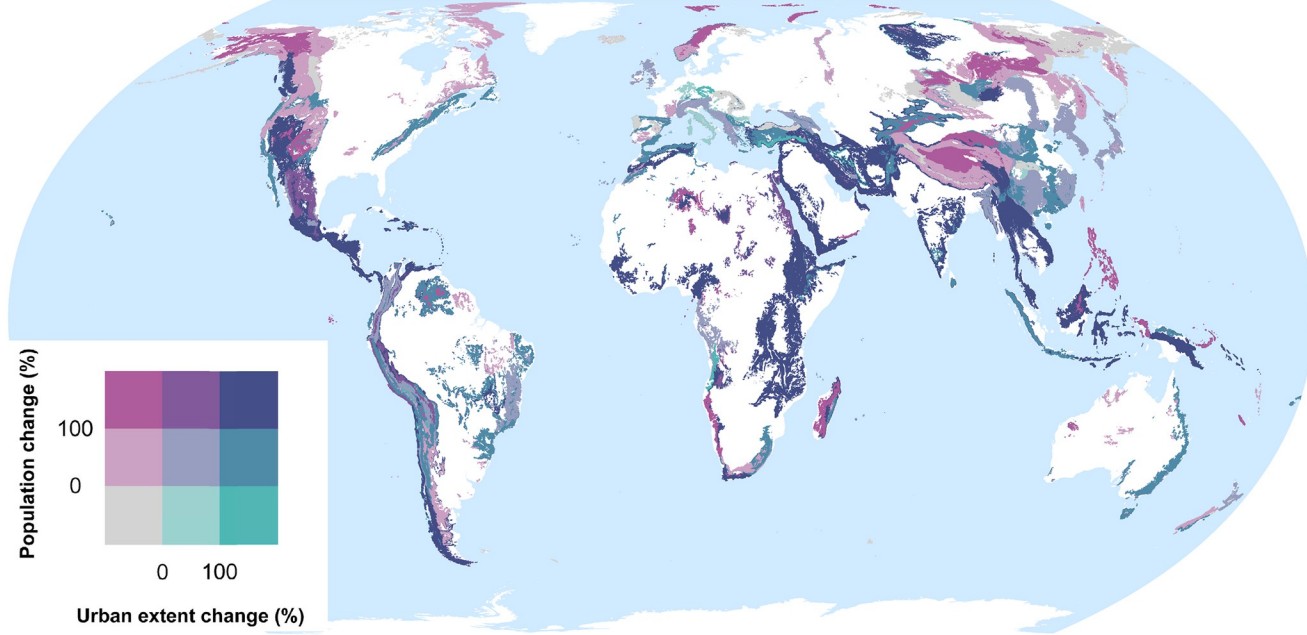

**Fig 3. Bivariate choropleth map of total population count change (%) vs. urban extent change (%) between 1975 and 2015 by GMBA sub-mountain range zone according to the GHS-POP and GHS-SMOD datasets.** Change values had to exceed the thresholds indicated to be placed in the higher category. Areas with declines in both metrics are shown in grey.

change) that is provided in the (S14 Fig in S1 File) presents a broadly similar pattern. The four underlying univariate maps are likewise provided in the (S11–S13 Figs in S1 File). As with all other datasets generated in this study, these output layers are also provided as vector files that can be readily ingested into various GIS software and web-mapping applications to aid potential re-use and/or further analysis.

## Regional aggregations

Next, we present the population count and density results that were computed and aggregated against the various relevant global spatial boundary datasets, including those used in high-profile global assessment exercises of the IPCC and the IPBES (see S4 Table in S1 File for a full list). In fact, the new population statistics generated by our workflow were recently reported in the Cross-Chapter Paper on Mountains in the IPCC's 6th Assessment Report [1, 63].

The numerous input combinations naturally generated a considerable quantity of output data, for which several presentation or visualization possibilities exist. Here, for illustrative purposes, we present a few selected example combinations. However, spatial population metrics for all input combinations—both counts and mean densities—are provided in the online S1 File. As such, similar maps to those presented here can be readily generated. The scripts provided would also enable the analyses to be repeated for other or new reporting datasets, such as the IPCC WGI regions.

Fig 4 reveals that the mountainous proportion of the population by IPCC regions (according to K1) is highest in Central and South America, and is lowest in Australia. The mountain population proportions of the other regions are fairly similar, between 11% and 19%, although at such a coarse aggregation level there could be substantial variations within IPCC regions that are not visually apparent.

Fig 5 shows the spatial pattern in the ratio of 2020 mountain population density to overall population density within IPBES areas according to K2. Whilst the mean population density

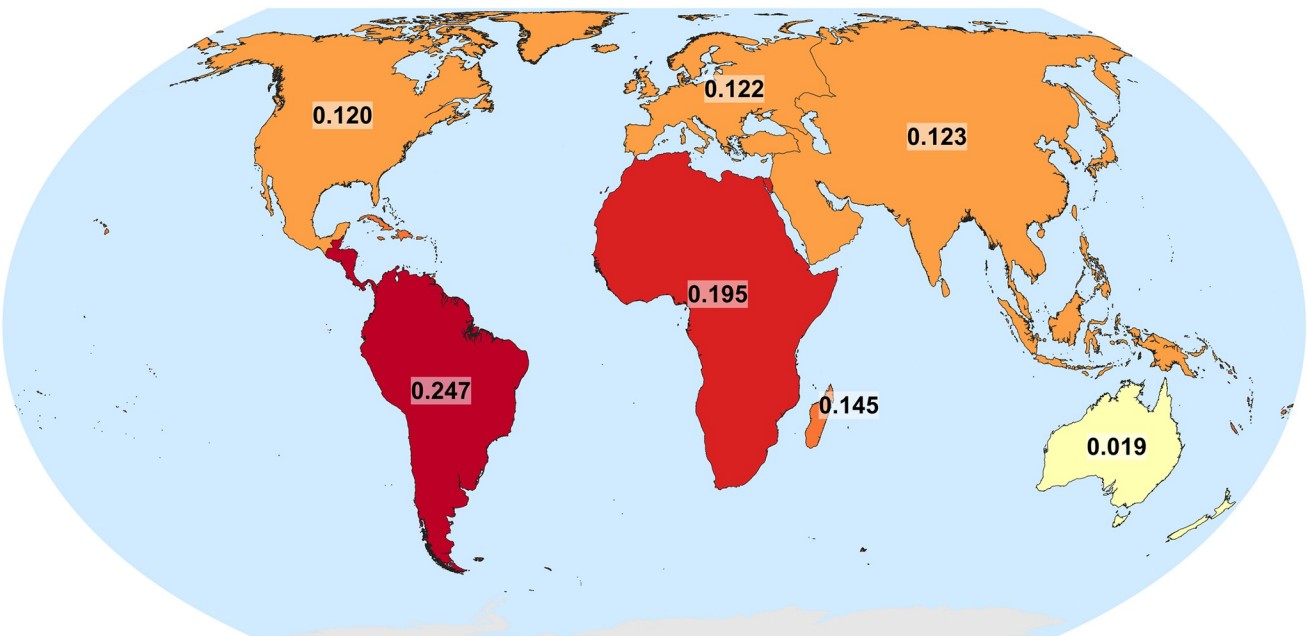

**Fig 4. Ratio of 2019 mountain-only population count to total population count by IPCC regions according to the K1 mountain delineation and LandScan population.** The 0.145 value corresponds to all "Small Island" regions globally.

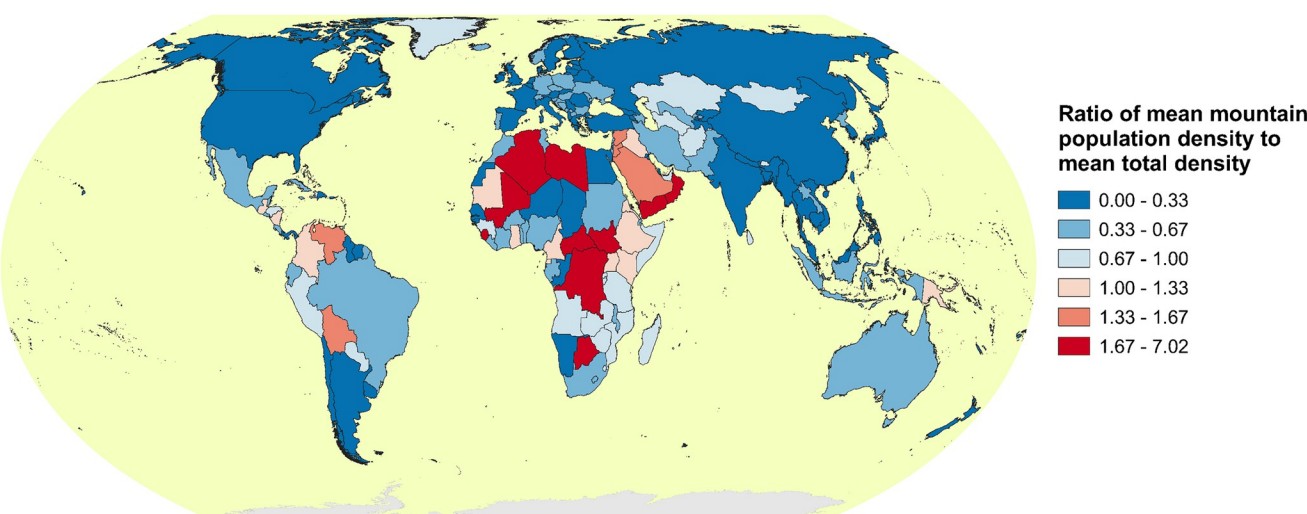

**Fig 5. Ratio of 2020 mean mountain-only population density to mean total population density by IPBES areas according to the K2 delineation and WorldPop population.** The 0.145 value corresponds to all "Small Island" regions globally.

in the mountainous portions of most areas is lower than the average densities (in broad alignment with the findings of Cohen and Small [41], and likely due to their cooler temperatures and related reduced productivity), in parts of South America and most especially Africa, this situation is reversed. Mountains are therefore clearly important human habitats in such regions. Enhanced precipitation associated with adiabatic cooling may be an important underlying mechanism.

From the output data provided, similar plots can be easily generated for the other reporting boundaries used. For instance, S19 Fig in S1 File shows the total population count of the BasinATLASv10 "level 6" hydrological catchments that contain at least some mountainous terrain according to one or more of the delineations. All such regional population breakdowns may help guide decisions regarding where (geographically) investments in research activities and other actions are likely to be most impactful.

### Influences on population distributions

Fig 6 shows comparisons between mean population density by entire World Climate Region (S8 Fig in S1 File) and only the mountainous parts thereof for 1975 and 2015. As such, it provides an initial indication of the extent to which climate and topography may influence population globally. For illustrative purposes, we only consider GHS-POP in conjunction with K1, although as noted above similar plots could be easily generated for other combinations using the outputs provided.

In most regions, both the total and mountain-only population density increased markedly between 1975 and 2015. In the majority of regions, the mountain population density in both years is lower than the total population density, in some cases substantially so (e.g. in Cool Temperate Moist, Warm Temperate Moist, and Tropical Moist regions). However, in some more arid zones (Cool Temperate Dry, Subtropical Dry, and Tropical Desert), the situation is reversed, with mean mountain population densities higher. This finding—that contemporary mountain population densities are generally highest in moist temperate and subtropical regions, where net primary productivity will be relatively high—accords with previous global findings [50, 51].

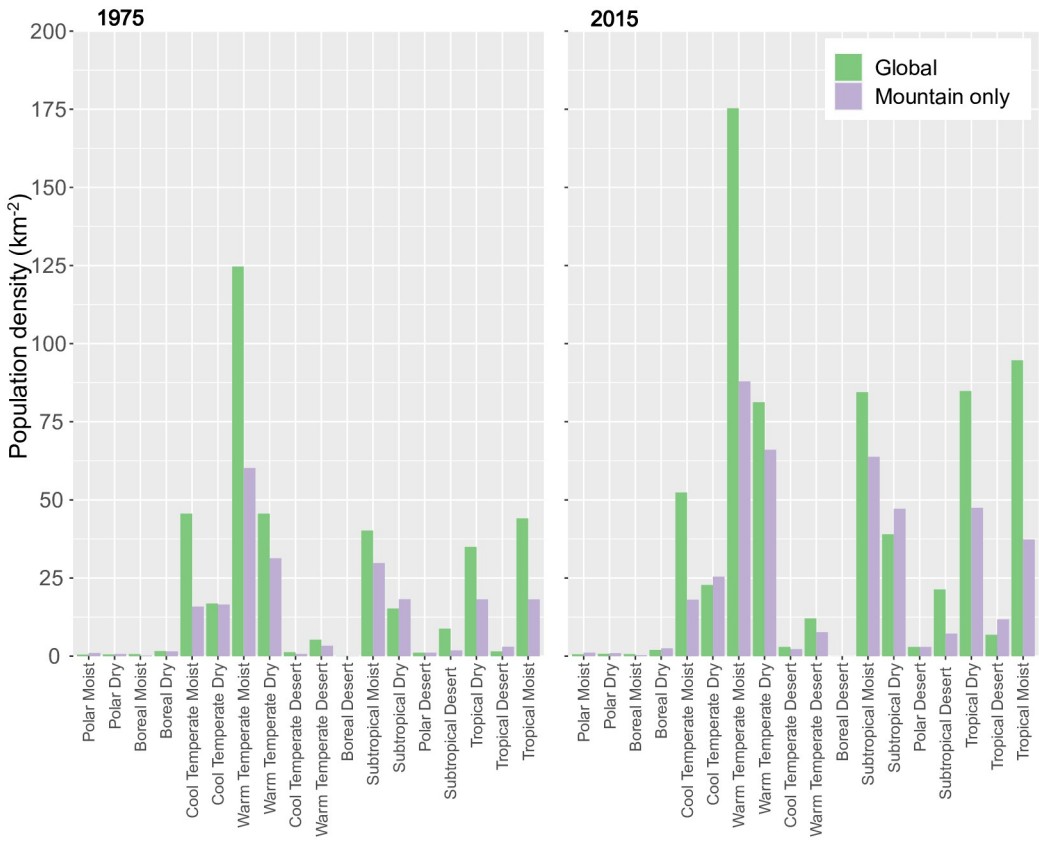

**Fig 6. Mean population density for entire World Climate Regions (WCRs) ("Global") and the mountainous parts thereof ("Mountain only") for 1975 and 2015 according to GHS-POP and the K1 mountain delineation.** Plots for the intermediate years (1990 and 2000) are included in an animated sequence in the online S1 File.

The ratio between total and mountain-only population appears to have widely remained relatively constant over time. This could indicate that whilst climate evidently exerts a strong influence on global population density, limiting conditions may have not yet been encountered. In the more arid regions, however, mountain population densities have increased more sharply than overall densities. This suggests that in such climates, the generally higher precipitation totals mentioned above (and perhaps also cooler temperatures) may allow mountain regions to act as important refugia for people.

Some caution must be exercised when interpreting these results, however. As mountains are already somewhat reflected in the spatial distribution of the overall climate regions (i.e. the presence of mountains dictates the distribution of climate zones in these areas), the respective influences of climate and topography cannot easily be distinguished in such plots. Furthermore, the climatic zones are temporally stationary, corresponding to the period 1970–2000 [60].

The more detailed investigation into the patterns of and potential controls on population distributions, which took a more spatially granular approach on account of the strong environmental (predominately topography-driven) gradients that are characteristic of mountain regions, to provide further insights. This component of the analysis looked *within* individual mountain regions (i.e at sub-mountain range scale). From *gmba_v2_0_k_union_diss_contours_covariates_joined.sqlite* and *gmba_v2_0_k_union_diss_contours_covar_joined_for_*

*matrix_plt.sqlite*, the mountain range sub-zones that are the most populous, have the highest population densities, are the most urbanized, are the most protected (proportionally), have experienced the most pronounced population, urban extent and/or urban populations changes over recent decades, and are climatologically the hottest, coldest, wettest, or driest can be rapidly identified.

The correlation matrix (Fig 7) summarizes the relationships between all variable pairs in the pooled dataset (i.e. the data attributed to all mountain range sub-zones). A more detailed version of this matrix is provided in the (S15 Fig in S1 File). The plots corresponding to the future population projections (SSPs for 2050 and 2080) are provided only in the online S1 File.

The directions of these relationships between the population density metrics (*p.den.1975* and *p.den.2015*) and other covariates align with expectations. For instance, positive associations with both annual mean temperature (*ann.mean.t*) and the temperature of the coldest quarter (*t.coldest.q*) are observed, indicating that warmer mountain temperatures are generally associated with denser populations. Seasonality of temperature (*t.seas*), meanwhile, is negatively associated with population density; mountain climates with high interannual temperature variability thus appear to be less conducive to dense human populations. Higher mean annual precipitation (*p.ann*) is also associated with higher population density in general. The negative relationship with *p.driest.quarter* (generally higher population densities where drier) could appear counter-intuitive at first, but may reflect the need for relatively dry growing conditions (or more specifically an optimal "corridor" in terms of moisture conditions).

Again expectedly, relationships with the topographic variables are negative, meaning that increasing mean elevation (*elev*), slope angles (*slo*) and topographic roughness index values (*tri*) are associated with generally decreasing population densities, although these correlations are weaker—and perhaps more so than may have been anticipated. Similarly, the protected-area proportion (*prop.prot.2021*) is negatively associated with mean population density, although again weakly. The direction of this association with protected areas is, however, consistent with the findings of the large, global-scale meta-analysis of Luck et al. [49], suggesting that higher human population densities affect the amount of land set aside for protection of conservation in mountains as well as other settings (see also [64]).

The single covariate with the highest correlation coefficient (strongest relationship with population density) with mean population density is *ann.mean.t*. Overall, several climate variables demonstrate stronger correlations than the topographic variables that were considered, with temperature variables generally appearing more influential than precipitation. For instance, the seasonality of precipitation (*p.seas*) and precipitation during the driest quarter (*p.driest.q*) are only weakly (and sometimes even insignificantly) associated with population density. These relatively weak associations with precipitation could be considered somewhat surprising given the apparent link between population density and aridity suggested by Fig 6. More expectedly, strong positive relationships are apparent between the proportion of the mountain regions that can be considered urban and mean population densities across those zones.

S8 Table in S1 File summarizes the correlation coefficients between the population variables and potential covariates numerically, now also including the future population scenarios mentioned above. As such, it illustrates the extent to which relationships between the population density metrics and potential covariates considered have varied historically, and to what extent these relationships are expected to be maintained or evolve according to the latest (global) population projection scenarios. Looking over the historical period, for which we can have greater confidence, the climatic dependency seems to have strengthened; specifically, correlation coefficients are stronger in 2015 than 1975. That climate is assumed stationary in this analysis is an important caveat, however.

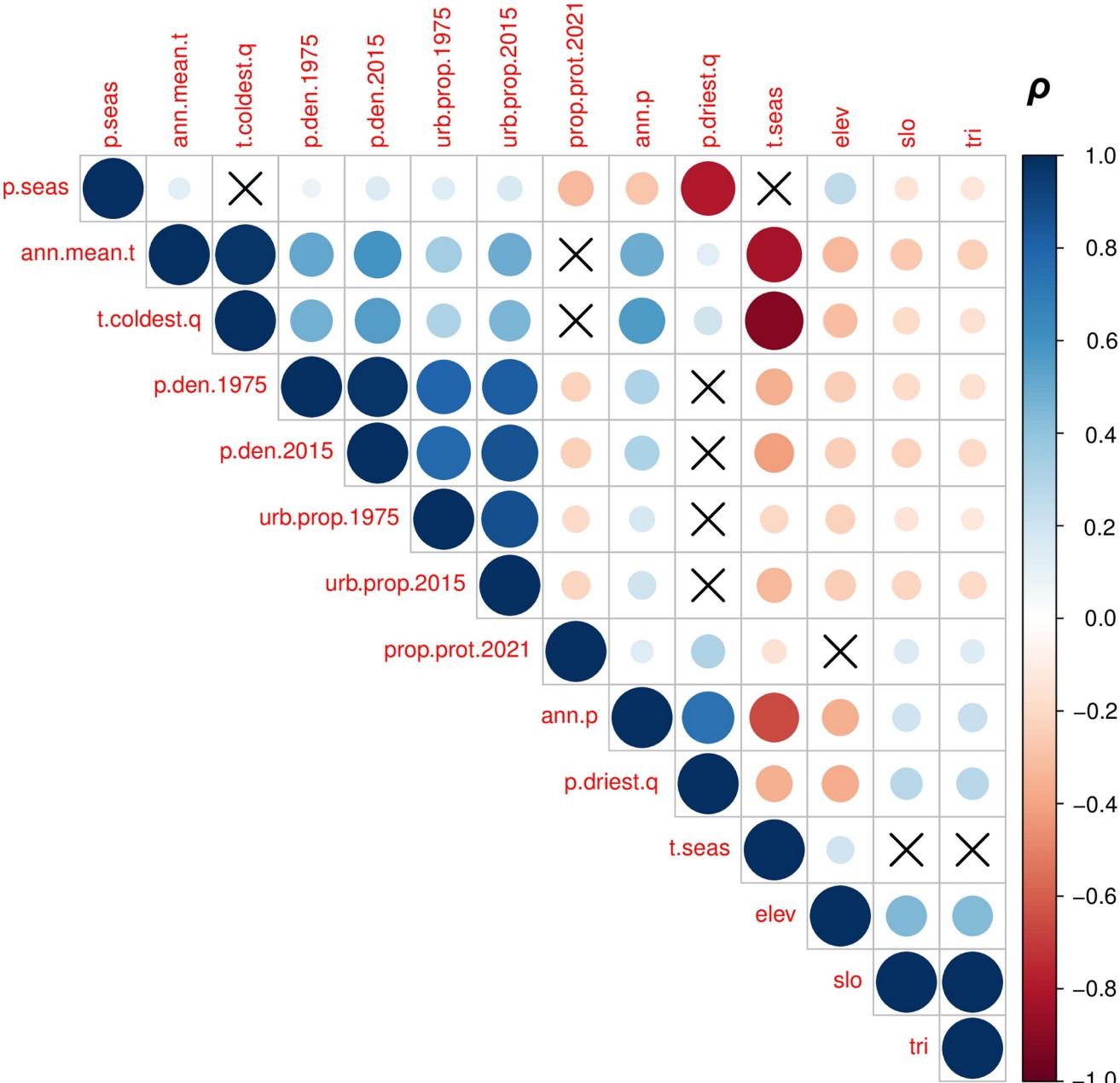

**Fig 7. Spearman's Rank Correlation matrix of population and spatially averaged potential covariate data across sub-divided "Low", "Middle", and "High" GMBA mountain polygons for the historical period.** *p.seas* denotes mean precipitation seasonality according to WorldClim v2.1, *ann.mean.t* annual mean temperature according to WorldClim v2.1, *t.coldest.q* the mean temperature of the coldest quarter according to WorldClim v2.1, *p.den.1975* population density in 1975 according to GHS-POP (a "dependent variable"), *p.den.2015* population density in 2015 according to GHS-POP (a second "dependent variable"), *urb.prop.1975* the areal proportion of each zone considered urban in 1975 according to GHS-SMOD, *urb.prop.2015* the areal proportion of each zone considered urban in 2015 according to GHS-SMOD, *prop.prot.2021* the areal proportion of each zone considered protected according to the May 2021 release of the World Database on Protected Areas (WDPA), *ann.p mean* annual (sum) precipitation according to WorldClim v2.1, *p.driest.q* mean precipitation (sum) of the driest quarter according to WorldClim v2.1, *elev* is mean elevation according to the MERIT terrain model, *slo* is mean slope angle according to the Geomorpho90 dataset, and *tri* is topographic roughness index according to the Geomorpho90 dataset (see S5 Table in S1 File for further details on these datasets). A value of +1 indicates a perfect positive correlation, whilst -1 indicates a perfect negative correlation (sphere size is also proportional to relationship strength). Relationships that are statistically insignificant at the 0.01 confidence level are indicated with crosses.

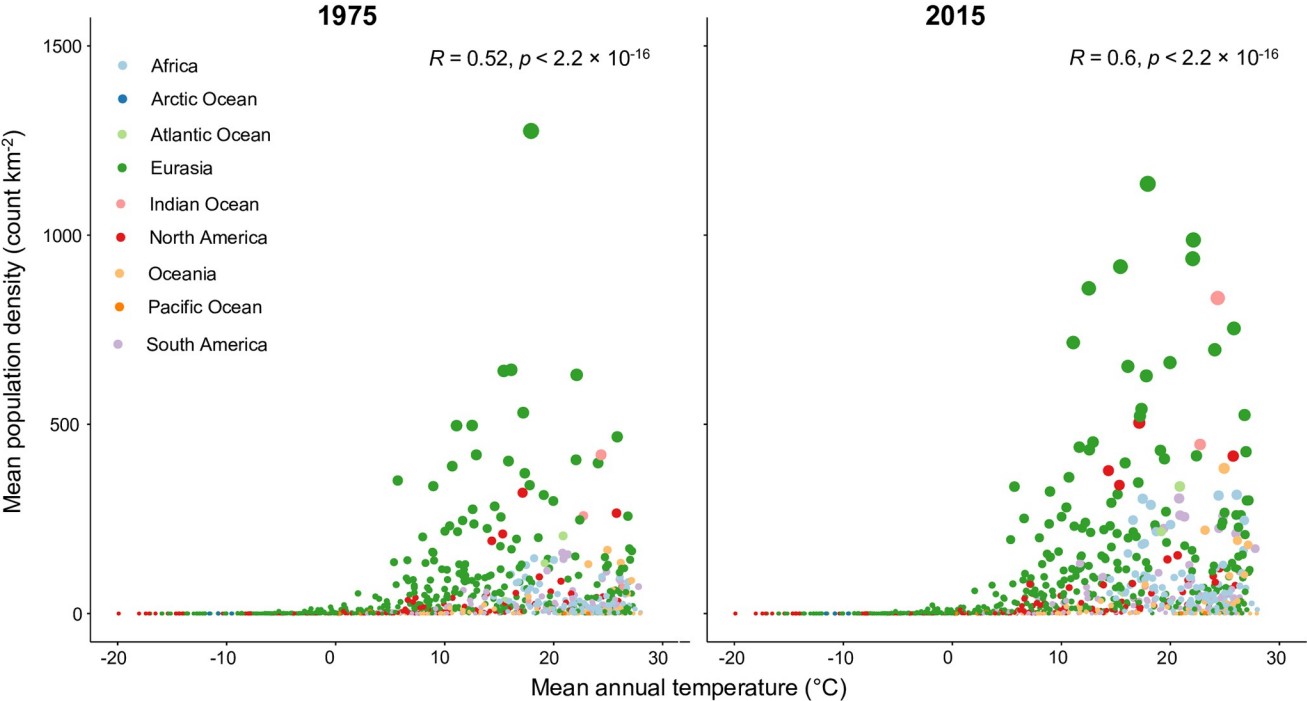

**Fig 8. Bivariate scatter plots of mean population density against mean annual temperature by GMBA mountain sub-zone for 1975 and 2015.** Note that the climate is considered stationary in the present analysis. Note also that the regions are those indicated in the GMBA Mountain Inventory, which are different from those used so far because some mountain ranges span the boundaries represented in the other reporting datasets. "Oceans" refer to ocean island mountains. Scatter plots for numerous other variable pairs are provided in the online S1 File.

The associations can be further assessed by means of bivariate scatter plots. Fig 8 presents an example of such a plot for the variable with the highest correlation coefficient (strongest correlation with population density), mean annual temperature (*ann.mean.t*). This plot shows that the relationship between human population density and temperature is highly non-linear. It also again reveals that mean mountain population density has generally increased in most regions, but that this has occurred predominantly in areas with more benign (i.e. warmer, but not extremely hot) conditions; perhaps corresponding to humans' thermoneutral zone, in which the need for additional heating or cooling systems is minimized. Alternatively (or as well), that the strongest increases have been in these warm zones may indicate that net primary productivity generally outweighs pathogen stress (both increase with temperature; [51]) under these conditions, with pathogens perhaps representing a stronger limiting factor in hotter zones (and especially high density cities with poor sanitation and healthcare systems therein). In colder settings, especially where mean annual temperatures are below approximately 5°C, little change has occurred. Whilst the highest population densities for both time periods are associated with moderate temperatures, no clear upper temperature limit on mountain population can be discerned. Such upper limits were not specifically investigated, however, e.g. mean temperature of the warmest quarter was not considered as a potential covariate here, although this could easily have been done using the same approach as for the others. In theory, such upper limits could affect mountain populations, both directly (e.g. if temperatures in already warm large cities in or very near mountains become unbearably hot with climate change), or else changing climatic conditions in more distant areas (e.g. leading to failures of agriculture [65], post-"peak water" [66], or again simply more prolonged occurrence of

temperatures that are too hot for long-term human survival) drive migration into more temperate mountainous areas (see also Luck et al. [49]).

Many other relevant pairwise scatter plots are provided in the online S1 File. Most exhibit a high degree of variability, demonstrating that both high and low population densities can occur under a range of conditions, although as in Fig 8 stronger constraints are evident towards the extremes of the covariate distributions in many cases. Total population metrics and urban proportions for the corresponding years (not shown here) are, unsurprisingly, closely correlated. This further reaffirms the often strong link between mountain population growth and urbanization exemplified by Fig 2. North America is something of an outlier, however, in that mean population density across its mountainous sub-zones increases much less sharply with increasing urban proportion than elsewhere. This seemingly highlights the relatively low density of urban areas in North American mountains.

Based on these quantified past relationships, and similarly to traditional plant and animal species distribution models [67], it could be possible to statistically model the expected spatial response of human populations in mountains to plausible future conditions. However, the omission of certain key processes, dynamics, and interactions in such models can compromise their results [68, 69]. Thus, reliably predicting human populations in mountains would certainly require that the multi-faceted interplay between natural, social, and economic phenomena and characteristics, including any regional and temporal variability thereof, be incorporated.

Another interesting aspect of S8 Table in S1 File with regards to the future is that—if one makes the (strong) assumption that future mountain climate shifts will be relatively modest compared with population shifts and can therefore be temporarily neglected—the dependencies between population densities and this stationary climate are expected to strengthen by 2050 and 2080 under the future population projections (SSPs). This could be a reflection of the expectation in the scenarios that most future mountain population growth will occur in regions which, historically, have been climatologically benign, and/or that population declines may occur preferentially in regions whose climates have historically been harsher, as Fig 6 suggests has been occurring. Alternatively, it could be purely coincidental (e.g. if current climate is not accounted for directly in the future SSP total population projections). Extending this exercise to assess the extent to which historical correlations are maintained in joint future climate and population projections should be a priority. In contrast, dependencies on topographic variables (which can be considered essentially temporally stationary on the timescales in question) are expected to weaken under the future projections. This may indicate that mountain-specific topographic constraints on population distributions are inadequately accounted for in the current generation of global population projections.

To summarize this sub-section, whilst our analysis has generated much data (presented both here and in the online S1 File), we are not yet in a position to make strong statements about many of the possible associations that have emerged. Above all, this applies to the part of our analysis dealing with future population projections, which remains highly speculative. We therefore invite other researchers to analyze the data presented more thoroughly and extend our work where necessary.

## Limitations

Our study is associated with several limitations. Firstly, whilst individual mountains can be easily identified and mapped as discrete landform entities, and multiple contiguous mountains without intervening valleys are also easily recognized as discrete mountain ranges, across larger areas the distribution of mountains is often more scattered and/or mountains are separated by intermontane valleys of variable widths. All established delineation approaches

identify such areas as mountain regions despite them containing non-mountain landforms like hills, valleys, and foreland. In other words, all delineations used here inevitably include some terrain that is strictly non-mountainous, but do so to different extents. This is important because it is this non-mountainous terrain that often becomes urbanized. The unavoidable ambiguity explains our insistence that the expression "mountain (urban) population" should actually be taken to mean "(urban) populations living in and near mountainous regions". On a similar theme, we did not investigate global patterns in the extent to which mountain population estimates according to the various input combinations (dis)agree with each other. This could be done straightforwardly using the data provided, however.

Notwithstanding the above, by providing estimates mainly on the populations living "in" (or very near) mountains themselves, we did not quantify the populations in connected lowland regions that currently benefit from mountain ecosystem services, and who may therefore be exposed to any adverse impacts of mountain system change. That said, at least for water supplies, this question has already begun to be addressed [23, 70]. Still, better characterising the spatial "degree of connectedness" between mountains and lowland populations with respect to both water (especially in terms of groundwater recharge [71] and inter-basin transfers) and other ecosystem goods and services would be critical.

The gridded population datasets used are also subject to uncertainty, much of it related to the methods by which census data are disaggregated. For instance, GHS-POP uses the built-up density per grid cell, which is in turn derived by processing optical satellite imagery. The accuracy of the resultant outputs therefore depends on image quality as well as the processing techniques applied. Unfortunately, none of the population sources employed here provide any information on the probability distribution around the central estimate for each cell. For this reason, population grid uncertainty for a given mountain delineation could only be considered according to the spread of the "ensemble" for the common year of 2015.

Regarding the more explanatory component of the analysis, numerous additional factors that could strongly influence mountain populations were neglected to keep the scope manageable. In particular, looking ahead, it could prove fruitful to include socio-economic and demographic variables like birth rates and age distributions, data on healthcare and education, transport and digital connectivity, economic conditions for agriculture and other activities, levels of governance and outside investment, efficiency of responses to past natural hazard events, and the perception of efforts to mitigate the impacts of future events. Obtaining sufficiently comprehensive and standardized data (even across individual mountain regions, never mind globally) on such factors such that they could also be incorporated remains a major challenge. This is a priority area for GEO Mountains—an Initiative of the Group on Earth Observations (GEO) seeking to improve mountain data discoverability, accessibility, and usability [72]—to address in collaboration with both experts and local stakeholders.

Finally, the introduction of future climate projections to address possible dependencies between projected future climate and population patterns in mountains fell beyond our scope. Given the likelihood that change patterns will vary considerably between regions and may furthermore be elevationally differentiated [73–76], this is not ideal. That said, applying climate models in mountains remains challenging; spatial resolution is extremely coarse relative to the variability of processes of interests, ensembles exhibit considerable disagreements, and important biases are present. Certain variables (e.g. precipitation) are more problematic than others.

## Outlook

Reliable estimates at relevant spatial scales on the present-day distribution of human populations living in and near mountains represent a key element of most attempts to assess the

impacts of changing ecosystem services and natural hazard frequency-magnitude relationships upon mountain societies in a robust and holistic fashion [77]. Such population data are also critical to ensure that any subsequent interventions related to environmental management and protection, climate change adaptation, natural disaster mitigation and response, and sustainable development and aid provisions are adequately, equitably, and proportionally focused and resourced (e.g. where rapidly changing and often extremely hazard prone regions, or where water supplies are especially threatened by climate change, coincide with and large populations). As such, in future work, breaking down the population estimates by additional demographic variables such as age and sex could also prove useful in many ways, for instance in assessing vulnerability to change, identifying the most appropriate aid and intervention strategies [78], and better projecting future population dynamics.

It may also be noted that most previous (global) studies exploring correlations with human population employed more direct drivers, such as net primary productivity, biodiversity, and pathogen stress as covariates, rather than the more indirect environmental conditions considered here (these more direct drivers are nevertheless intrinsically linked to environmental conditions; [50]). It could therefore be prudent for future mountain studies to use such more direct covariates explicitly.

In addition, as already mentioned, improved future projections of mountain population distributions under various possible climate and broader socio-economic scenarios could also be informed by first exploring the extent to which the dependencies between climate and other environmental covariates that have been identified here over the recent past are maintained when existing future population and climate scenarios are considered simultaneously. As part of this, it will likely be necessary to critically assess the extent to which the patterns in mountainous areas embedded in the best gridded population projections currently available—the (global) SSP projections—are reflective of, or are meaningful in, the specific contexts of these regions. The inherently high uncertainty in climate projections in mountains should also be accounted for. At present, mountain topography is understood to not be considered.

Slightly extending the workflow would also enable additional questions to be addressed. For instance, links between mountain population change and land cover change over recent decades could be explored, and the human population at risk from certain natural hazards could be quantified. Key data gaps or limitations would have to be overcome to enable this, however. For instance, with respect to flooding, which is generally recognised as one of the most dangerous, costly, and relatively frequent natural hazards (in mountains but also globally), while high-resolution population count (see above) and other exposure datasets (e.g. physical asset values; Eberenz et al. [79]) are increasingly available, improved hazard data are required by mountain researchers and practitioners globally.

According to Aerts et al. [80], the only available open-source flood hazard datasets have rather coarse spatial resolutions, neglect pluvial flooding and, crucially, only cover large catchments whose areas exceed certain predefined thresholds. None of these attributes are ideal in mountainous terrain. In contrast, private sector providers such as JBA Risk Management Limited do provide flood hazard maps for both pluvial and flooding at much higher spatial resolution that include the effects of flood defenses (where known) and employ no minimum catchment area threshold. A way forward may thus involve reaching agreements by which such data could be accessed and applied more widely for humanitarian and related purposes. Accounting for climate change-driven hazard non-stationarity [81] represents an ongoing challenge even for these providers, however. Building catalogues of remotely sensed flood extents [82] represents an alternative approach, but the full range of possible events in time and space will not have occurred during the satellite era, and key hazard metrics like water depth and velocity are difficult to derive from such sources.

Other mountain hazards (e.g. avalanches, lake outburst floods, earthquakes, and landslides, which are often cascading or compound events) are arguably even more challenging to quantify probabilistically, not least because they are more highly non-stationary under changing climatic conditions. Yet without such information, effective risk mitigation and transfer mechanisms cannot be implemented. Our sights must be set ambitiously on developing coupled or integrated projections of future mountain ecosystems and their associated services, natural hazards, and human populations at meaningful spatio-temporal resolutions. If achievable, such simulation capabilities would revolutionise mountain climate change adaptation and environmental protection measures. Increasing the availability of mountain-specific hazard and exposure data and models available via initiatives such as the Oasis Loss Modelling Framework could be beneficial in this regard.

More generally, reproducibility is—or at least should be—a central tenet of the scientific process. However, many recent studies across a range of scientific disciplines have been found to be irreproducible, which poses a major challenge the credibility of science [83–85]. The chances of a given workflow or analysis being reproducible are greatly enhanced if the precise datasets employed are clearly stated, are findable and interoperable [86], full codes and/or algorithmic details are shared, and said algorithms are implemented in open source software (again with details such as version numbers indicated). Although these standards are not necessarily always easy to attain, we must endeavor to do so. For analyses involving geospatial operations specifically, the continual maturation of community-developed software such as GDAL, R, QGIS, PostGIS, and Python, underpinned by the gathering momentum of the Open Science movement more generally, make the implementation of the "transparent and reproducible" philosophy not only possible, but often highly in terms performance when compared with traditional Graphical User Interface (GUI)-based applications.

## Conclusions

We developed and implemented an open and reproducible workflow to characterize the spatio-temporal distribution of human populations living in and near the world's mountains over recent decades. Compared with previous efforts that only provided populations estimates for more limited input dataset combinations, and moreover often applied rather opaque, non-reproducible methods (e.g. omitted details of the specific datasets and algorithms used), our approach is considerably more standardized, systematic, comparative, and transparent in nature.

This is important because, as in countless other fields, the ability of data users to develop a sound appreciation of the likely impacts of alternative input dataset choices on any eventual outcomes or conclusions is paramount to sound subsequent decision making and actions. Only comparative analyses such as that presented here permit this. Additionally, by introducing several additional datasets on topography, climate, urban extents, and protected areas, we explored the extent to which these factors influence human population distributions within individual mountain regions for the first time. Our main findings are that:

- Variability in mountain population estimates is dominated by the choice of mountain delineation. Population dataset choice is not negligible, however, especially for smaller spatial analysis units; for instance, substituting GHS-POP for WorldPop causes a change of up to 39.9% in the estimated global urban mountain population, all else being equal (Fig 1 and Table 1);

- In many mountain regions, population increases over recent decades have been associated with strong urbanization in both extent and population, although population and urbanization trends are disconnected in some regions (Figs 2 and 3);

- In parts of Africa especially, mean population densities in mountainous regions are notably higher than densities more generally. This suggests that, broadly speaking, mountains provide important "refugia" for human populations in certain dry and/or hot climates (Figs 4–6);

- At sub-mountain range scale (i.e. within individual mountain regions), moderate and high mountain population densities are found to occur under a relatively wide range of climatological and topographic conditions (Figs 7 and 8, S8 Table in S1 File). That said, some evidence of climatic controls, especially measures of temperature, exists. It certainly appears that climatic variables may generally exert a stronger influence on mountain densities than topographic variables, at least at the spatial scales considered. Moreover, in many instances, these dependencies appear to have strengthened through time as population growth has preferentially occurred in regions with more favorable conditions (S8 Table and in S1 File). Overall, these findings align with those of previous, global studies that have analysed the influence of more specific population covariates for both pre- and post-agricultural/industrial societies, and;

- Correlations between population metrics and potential covariates evident over the historical period are not consistently maintained when future global population projections are substituted (S8 Table in S1 File); this could indicate either that these influences break down under future conditions, or else that future population projections somewhat overlook the specific factors that affect mountain social-economic systems and populations. Further work in this area is required, however.

Studies presenting basic summaries of the number of people living within a few meters of the present-day coastline globally arguably revolutionized, and continued to reinforce, our collective understanding of the potential impacts of ongoing sea-level rise [87]. Somewhat similarly, the results presented here demonstrate that irrespective of the specific combinations of input datasets taken, a considerable proportion of the Earth's population live in or near its mountains, and may therefore be increasingly "squeezed" by ongoing changes at both higher and lower elevations. Conversely, if environmental protections and management strategies are insufficient or ineffective, these populations may themselves leave increasingly indelible imprints upon highly valuable mountain environments and ecosystems.

In an era when the credibility of science is being questioned in certain quarters, we should collectively aspire to deliver methodologies are as transparent and reproducible as possible. Here, with a view to minimizing the science-policy gap and, hopefully, generating relevant datasets with high reuse potential—especially for applications such as the IPCC and IPBES assessments—such an approach was implemented. Our results were aggregated against several relevant reporting boundaries, and are provided in an easily usable spatial format. As such, they can easily be visualized and, if desired, further processed in standard desktop GIS applications. In this way, our research will hopefully translate into more informed and robust policy decisions related to urgent issues like environmental management and protection, sustainable development, risk mitigation, and climate change adaptation. Our insights into the associations between population and potential covariates may also contribute to improved predictions of mountain population distributions under plausible future climate and broader socio-economic and demographic change scenarios.

In summary, we provide a sound and objective basis for various important subsequent applications involving people and mountains across traditional disciplinary boundaries and spatial scales. We recommended that users of mountain population estimates engage more closely with the various underlying methods and data choices to develop a fuller appreciation

of how conclusions and ultimately important decisions are made. We see great potential for the datasets generated by our fairly comprehensive, comparative analysis to be further exploited and/or developed. Finally, the scripts provided may also prove useful for spatial analyses in other disciplines involving similarly large and varied spatial datasets.

## Supporting information

**S1 File. Please see the attached SI File.** In addition, input and output data, code, and high-resolution and supplementary figures are available at: doi.org/10.5281/zenodo.6673651. Spatial outputs (.sqlite) can be easily visualized in QGIS (simply drag and drop).
(PDF)

## Acknowledgments

We are extremely grateful to the developers and maintainers of the numerous open datasets and open-source software packages used. We thank Mr. Matthias Fries for IT support, Prof. C. Körner for useful discussions, and Dr. E. Beever for thoughtful comments on an earlier draft. Any use of trade, firm, or product names is for descriptive purposes only and does not imply endorsement by the U.S. Government.

## Author Contributions

**Conceptualization:** James M. Thornton, Mark A. Snethlage, Philippus Wester, Carolina Adler.

**Data curation:** James M. Thornton, Roger Sayre, Daniele Ehrlich.

**Formal analysis:** James M. Thornton.

**Funding acquisition:** Philippus Wester, Carolina Adler.

**Investigation:** James M. Thornton.

**Methodology:** James M. Thornton, Mark A. Snethlage.

**Project administration:** Carolina Adler.

**Resources:** Carolina Adler.

**Software:** James M. Thornton.

**Supervision:** Roger Sayre, Davnah R. Urbach, Daniele Ehrlich, Philippus Wester, Carolina Adler.

**Validation:** Daniel Viviroli.

**Visualization:** James M. Thornton.

**Writing – original draft:** James M. Thornton, Roger Sayre, Davnah R. Urbach.

**Writing – review & editing:** James M. Thornton, Mark A. Snethlage, Roger Sayre, Davnah R. Urbach, Daniel Viviroli, Daniele Ehrlich, Veruska Muccione, Philippus Wester, Gregory Insarov, Carolina Adler.

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
