## [Decision Letter · Decision Letter 0]

19 May 2022

PONE-D-21-39676Human populations in the world’s mountains: spatio-temporal patterns and potential controlsPLOS ONE

Dear Dr. Thornton,

Thank you for submitting your manuscript to PLOS ONE. After careful consideration, we feel that it has merit but does not fully meet PLOS ONE’s publication criteria as it currently stands. Therefore, we invite you to submit a revised version of the manuscript that addresses the points raised during the review process.

ACADEMIC EDITOR: This is an excellent, well thought out study that presents interesting results and acknowledges, as well as quantifies the limitations of the data and methodologies used. The paper nearly meets Plos One criteria for publication. There is a large literature on the ecological/climate and social-economic correlates of human population density. The paper should better integrate with this literature and, I think, would be strengthened by acknowledging the correspondence of results across studies of global variation in human population density. My full review is appended below. I wish the authors the best with their revisions and interesting research.

Summary: This paper is an exploratory description and analysis of human settlement and density in mountainous regions. The authors quantify mountain population distributions using open source data and methodology. They also assess the robustness of their descriptions to changes in the data sets used, in particular data sets for estimating mountainous areas (K1,K2,and K3), as well as global estimates of human population. Finally, the authors identify covariates of mountain population density—like temperature. The authors demonstrate that the choice of data set, and the underlying methodology used to construct the data set, changes the results of their calculation of human population density and urban settlement. Further, the authors conclude that human population density in mountains correlates most strongly with temperature. The results help build a foundation for modeling the effects of climate change on the distribution of humans in mountains.

Substantive Comments:

(1) The authors could and should integrate their research better with previous work on the ecological and social-technological determinants of human population density. See references below. Although this work does not focus on mountains specifically, it does provide previous results relevant for predicting the factors that impact human population density in mountains. Given the close correspondence of the current paper’s results and the results of these previous studies, it seems worth noting that similar climate (ecological factors) associate with human population density, regardless of whether the data are restricted to mountains only. Importantly, these non-linear relationships hold across different types of economic organization.  This redundancy of results across different data sets and study methodologies may signal an important fundamental result useful for modeling human responses to climate change.

Minor Comments/reading Notes:

(1) Abstract should not exceed 300 words. I did not count the words, but it reads long. Please double check

(2) “likewise responding to

evolving conditions.” Change `evolving’ to ‘changed’ conditions

(3) ``Elucidating any such relationships could lead to improved understanding of the

underlying drivers, and could even enable the identification of topographic, climatic,

and other conditions under which high population densities can develop and be

sustained in mountains.”

See references below on global investigations of the climate, ecology, and technological covariates of human population density:  These should be reasonable places to start to understand the processes underlying variation in human population density. The results are relevant, though none focus on mountains specifically.

@article{Freemanetal2020,

  title={The global ecology of human population density and interpreting changes in paleo-population density},

  author={Freeman, Jacob and Robinson, Erick and Beckman, Noelle G and Bird, Darcy and Baggio, Jacopo A and Anderies, John M},

  journal={Journal of Archaeological Science},

  volume={120},

  pages={105168},

  year={2020},

  publisher={Elsevier}

}

@article{Tallavaara2018,

  title={Productivity, biodiversity, and pathogens influence the global hunter-gatherer population density},

  author={Tallavaara, Miikka and Eronen, Jussi T and Luoto, Miska},

  journal={Proceedings of the National Academy of Sciences},

  volume={115},

  pages={1232--1237},

  year={2018},

  publisher={National Acad Sciences}

}

@article{luck2007review,

  title={A review of the relationships between human population density and biodiversity},

  author={Luck, Gary W},

  journal={Biological Reviews},

  volume={82},

  number={4},

  pages={607--645},

  year={2007},

  publisher={Wiley Online Library}

}

(4) ``these outputs are also provided are

provided as vector files”

 Typo.

(5) ``However, in some more arid zones (Cool Temperate Dry, Subtropical Dry, and Tropical

Desert), the situation is reversed, with mean mountain population densities higher.”

I think that this observation fits with first principles of food availability and disease as important factors that impact human population density. In high latitude areas, mountains are cold and less productive. In dry temperate and dry tropical areas, mountains often receive more rainfall due to adiabatic cooling and provide an ``escape” from pathogen vectors.

(6) Important to note that the relationship between temperature and population density is non-linear. This may be due to an optimal thermal zone; however, the authors should see the papers cited above. At a global scale, net primary productivity and pathogen stress correlate with temperature. So, as productivity increases so does pathogen load. In theory, there should be a point where the costs of living at higher densities, in terms of disease transmission mortality and morbidity, outweigh the benefits of more productive ecosystems.

We look forward to receiving your revised manuscript.

Kind regards,

Jacob Freeman

Academic Editor

PLOS ONE

Journal Requirements:

"We are extremely grateful to the developers and maintainers of the numerous open datasets and open-source software packages used. J.M.T, C.A., and P.W. acknowledge funding from the Swiss Agency for Development and Cooperation (SDC) (Project number: 7F-10208.01.02). We thank Mr. Matthias Fries for IT support, Prof. C. K¨orner for useful discussions, and Dr. E. Beever for thoughtful comments on an earlier draft."

"J.M.T, C.A., and P.W. acknowledge funding from the Swiss Agency for Development and Cooperation (SDC; https://www.eda.admin.ch/sdc) (Project number: 7F-10208.01.02).

GI acknowledges funding from the Institute of Geography, Russian Academy of Sciences (http://www.igras.ru/en/node/1) (Project number: 0148-2019-0007)

The sponsors played no direct role in the study design, data collection and analysis, decision to publish, or preparation of the manuscript."

Reviewers' comments:

Reviewer's Responses to Questions

**Comments to the Author**

1. Is the manuscript technically sound, and do the data support the conclusions?

Reviewer #1: Yes

2. Has the statistical analysis been performed appropriately and rigorously? 

Reviewer #1: Yes

3. Have the authors made all data underlying the findings in their manuscript fully available?

Reviewer #1: Yes

4. Is the manuscript presented in an intelligible fashion and written in standard English?

Reviewer #1: Yes

5. Review Comments to the Author

Reviewer #1: It is advisable

1. to specify abbreviations immediately after the first use of the term.

2. Revise and simplify the Materials and Methods

3. Check and correct the confusing lines eg. it becomes confusing While explaining K1, K2, and K3 and jumping directly to class 7 of K1 in last line of para 3 under Materials and Methods.

6. PLOS authors have the option to publish the peer review history of their article (what does this mean?). If published, this will include your full peer review and any attached files.

Reviewer #1: **Yes: **Dr MD. ASIF IQUBAL

---

## [Author Response · Author response to Decision Letter 0]

27 Jun 2022

Many thanks for your review. Please see the attached responses document. Kind regards, James Thornton.

---

## [Editor Report · Decision Letter 1]

1 Jul 2022

Human populations in the world’s mountains: spatio-temporal patterns and potential controls

PONE-D-21-39676R1

Dear Dr. Thornton,

We’re pleased to inform you that your manuscript has been judged scientifically suitable for publication and will be formally accepted for publication once it meets all outstanding technical requirements.

Kind regards,

Jacob Freeman

Academic Editor

PLOS ONE

Additional Editor Comments (optional):

Thank you for thoughtfully addressing all of the comments on the manuscript. Best of luck with your future research.